

# Modelling ultrafine particle growth based on flow tube reactor measurements

Michael S. Taylor Jr., Devon N. Higgins, and Murray V. Johnston

Department of Chemistry and Biochemistry, University of Delaware, Newark, Delaware, 19711, United States

*Correspondence to*: Murray V. Johnston (mvj@udel.edu)

**Abstract.** Flow tube reactors are often used to study the growth of secondary organic aerosol (SOA). Because a significant amount of growth must occur over the short residence time of the flow tube, precursor mixing ratios in a flow tube experiment are generally much higher than ambient values. In this study, a model of SOA growth based on condensation of nonvolatile molecules, partitioning of semivolatile molecules, and reaction of semivolatile molecules in the particle volume to produce

nonvolatile dimers, is used to compare particle growth under atmospherically relevant conditions to those under typical flow tube conditions. The focus is on the diameter growth of particles in the 10 to 100 nm diameter range, where growth rates can have a substantial impact on formation of cloud condensation nuclei. In this size range, both particle surface- and volume-limited kinetics may apply. Modelling shows that the higher precursor mixing ratios of a flow tube experiment cause surface-limited kinetics to be more prevalent in the flow tube than under atmospheric conditions. SOA formation is characterized by

the growth yield (GY), defined as the yield of oxidation products that are to grow the particles. Defined in this way, GY is the sum of all nonvolatile products that condensationally grow particles plus a portion of semivolatile particles that react in the particle volume to give nonvolatile dimers. Modelling shows that GY actually changes as a function of time within the flow tube. The experimentally determined GY from the measured inlet-outlet diameter change of particles in a flow tube experiment closely tracks the average of the time-dependent GY obtained from modelling specific chemical processes. Modelling is also

used to explore the effects of seed particle size (40, 60, 80 nm dia.), phase state (deliquesced vs. effloresced), and surface state (interfacial water), as well as precursor mixing ratio, all of which are shown to substantially influence SOA formation under the conditions studied.

## 1 Introduction (as Heading 1)

Atmospheric aerosols have significant effects on human health and the environment, from the direct inhalation of air into our

lungs to the changing composition of atmosphere (Najjar, 2011; Thompson, 2018). Particulate matter in the atmosphere has been a focus of attention since the London Smog Incident and similar events of the mid twentieth century (Bell et al., 2004). Both anthropogenic and biogenic emissions are major sources of new particle formation, whether it be through primary particle emissions or secondary formation from existing particulate matter (Després et al., 2012; Lehtipalo et al., 2018). Clusters of ambient molecules, such as ammonia, sulfuric acid, and organics with low volatility, are often sources for new particles in the



1 to 2 nm size range that are capable of spontaneously growing to larger sizes (Shrivastava et al., 2017). Once particles grow to the size range of ~100 nm, they can act as cloud condensation nuclei (CCN) which are capable of affecting radiative forcing on the earth (Johnson et al., 2018; Riipinen et al., 2011). Due to the many growth and removal processes involved in atmospheric particle growth, the likelihood of a nucleated particle to reach the CCN active range can vary greatly. Simulations of these processes contain large uncertainties with respect to secondary aerosols formed from biogenic emissions and the

varying properties of organic aerosols in general (Pierce and Adams, 2007).

Particle growth rates are generally studied in the laboratory using either chamber or flow tube reactors. With chamber reactors, precursors are added to the (sealed) reactor at time t=0 and the evolution of particles inside the reactor is monitored as a function of time (Pierce, 2017). With flow tube reactors, which are the focus of this study, precursors are mixed at the entrance

of the flow tube and the product distribution of aerosol exiting the tube is monitored (Krasnomowitz et al., 2019). Flow tubes provide the opportunity to quantitatively and comprehensively interrogate a specific time-point in the reaction, and the reaction conditions (start and end points) are precisely known and controlled (Stangl et al., 2019). The main drawback of a flow tube is that the time-point being studied is generally much shorter than that achieved in a batch reactor, necessitating somewhat higher precursor mixing ratios than are found in the atmosphere. In this study, we model particle growth experiments in a flow

tube reactor. The results provide insight into how best to interpret flow tube data and design meaningful experiments.

The focus of this study is particle growth by secondary organic aerosol (SOA) formation. SOA formation occurs when a volatile organic compound (VOC) is oxidized in the gas phase. There are usually a wide range of oxidation products from a given VOC precursor, and these products can be classified by their volatility, a measure of the product molecule's ability to

partition between the gas and particle phases or condense from the gas phase to particle phase. Low and Extremely Low Volatility Organic Compounds (LVOCs and ELVOCs) are able to condensationally grow particles, with LVOCs being limited by the Kelvin effect in small particles (<20 nm). Semivolatile organic compounds (SVOCs) are products with somewhat higher volatility, allowing them to partition between the gas and particle phases until an equilibrium is established. Oxidized volatile organic compounds (OVOCs) are products which are too volatile to partition to the particles, though they remain available to

participate in subsequent gas phase reactions (Bianchi et al., 2019). SVOCs and possibly OVOCs may contribute significantly to particle growth if they undergo multiphase reaction on the particle surface or within the particle volume to produce nonvolatile products that remain on/in the particle (Fuzzi et al., 2006; Gkatzelis et al., 2018).

SOA produced by the oxidation of biogenic volatile organic compounds (BVOCs) contributes significantly to fine particulate

matter present in the atmosphere (Jimenez et al., 2009). The molecular composition of biogenic SOA encompasses several hundreds to thousands of potential products that can be formed through various pathways, making its inclusion in atmospheric models complex (Heaton et al., 2009). Molecular analysis of biogenic SOA has shown evidence of particle-phase chemistry though detection of oligomers formed by accretion reactions, which enhance the uptake of organic matter into a particle over



what would be present from partitioning alone (Barsanti and Pankow, 2006; Tolocka et al., 2004). The reactivity of VOC
oxidation products can vary significantly, owing largely to the presence of one or more functional groups (Jia and Xu, 2020;
Zhou et al., 2018). For the oxidation of monoterpenes specifically, highly reactive hydroperoxide functionalities have been
found on up to 50 % of product molecules (Docherty et al., 2005; Mertes et al., 2012) which are thought to be formed by an
autooxidation mechanism (Bianchi et al., 2019; Crounse et al., 2013). Environmental factors such as relative humidity and
temperature affect oxidation product formation, especially with respect to product molecule reactivity and volatility, while
particle composition and phase state affect the multiphase processes these products may undergo (Zhang et al., 2015). By
studying these processes and properties, one is able to more accurately define and understand the lifecycle and effects of SOA
in climate cycles (Saha and Grieshop, 2016; Shrivastava et al., 2017).

SOA formation in chambers is generally quantified in terms of an aerosol mass yield, defined as the ratio of the mass
concentration of organic aerosol produced divided by the mass concentration of VOC consumed, and is interpreted based on
the partitioning and condensation of lower volatility products into the organic particle phase (Chen et al., 2011; Xavier et al.,
2019). The aerosol mass yield has the potential of being greater than one, since oxidation adds oxygen atoms to the molecule
and can give products whose molecular weights are greater than that of the original VOC. Aerosol mass yields for α-pinene
ozonolysis have been consistently measured and modelled at approximately 40 % under conditions of high mass concentrations
(Chen et al., 2011; Kristensen et al., 2017; Xavier et al., 2019). Aerosol mass yields have been shown to increase 2 to 4 times
upon the addition of seed particles, showing a significant dependence on seed size, type, and phase (Ahlberg et al., 2019). At
low mass concentrations, aerosol mass yields are generally independent of seed particle characteristics, as lower volatile
organics will condense while a relatively small fraction of semivolatile organics partition into the particle-phase. As the
particle-phase volume increases, the fraction of semivolatile organics that partition into the particle phase increase, simply due
to the shift in equilibrium where more particle volume is present to accommodate the additional semivolatile molecules
(Apsokardu and Johnston, 2018). While aerosol mass yield is useful for predicting the "end-state" SOA mass concentration, it
may be less useful for predicting the kinetics of SOA formation and hence the diameter growth rate of particles. Particle
diameter (or particle size distribution) is notably absent from the definition of aerosol mass yield, which only relates total
mass/volume of SOA produced to the amount of precursor reacted.


In this study, we introduce the term "growth yield" (GY) as a way of expressing how SOA causes diameter growth of particles.
GY, defined as the fraction of VOC oxidation events that gives a product able to enter the particle phase and stay there, causing
the particle to grow. GY is particle size dependent, since the impact of particle-phase chemistry relative to
condensation/partitioning increases in relative importance as the particle size increases (Apsokardu and Johnston, 2018).
Furthermore, GY is likely to be particle composition, phase, and morphology dependent, since these properties determine the
types of particle-phase reactions that can occur. Though not explicitly studied here, the Kelvin effect influences
condensation/partitioning of small particles (typically <<20 nm diameter) and can also contribute to the size dependence of



GY in this size range. As will be described later, GY is a key parameter that can be easily obtained by modelling the growth

of particles in a flow tube experiment. In this study, we use an enhanced particle growth model based on our original work

(Apsokardu and Johnston, 2018) to explore how GY changes with particle size, composition, and multiphase chemistry. The

model is first applied to growth of a single, isolated particle under atmospherically relevant conditions, and then to typical

conditions of a flow reactor (Krasnomowitz et al., 2019). The results give insight into how best to set up and interpret flow

tube experiments.

## 2 Modelling procedure (as Heading 1)

The growth model used in this study is built upon a foundation previously developed and described elsewhere (Apsokardu and

Johnston, 2018). Organic and inorganic species within this model include an ammonium sulphate seed particle, organic matter

of varying volatility, and water. Water can be present on the surface of a seed particle (where applicable) and in the gas phase

(humidity). The relative humidity for all simulations is maintained at a constant 60%, which is between the efflorescence

(~35%) and deliquescence (~82%) humidity points of ammonium sulphate (Gao et al., 2006). The modelling process begins

with an initial seed particle size that can be set at any diameter and treated as either an effloresced (solid phase) or deliquesced

(liquid phase) particle. Organic matter is distributed into six volatility bins which include one non-volatile organic compound

(NVOC), four semivolatile organic compounds (SVOCs), and one oxidized volatile organic compound (OVOC). Each organic

species has the potential to partition/condense between the gas and particle phases based on pre-set volatility parameters, as

discussed in Sect. 2.1. Multiple particle-phase reactions are incorporated into this model between the partitioned and condensed

organic matter and will be discussed further in Sect. 2.3. Water exists as a predetermined number of monolayers which cover

the surface of a solid seed particle when applicable. Previous studies have shown the presence of water on effloresced seed

particles under conditions of relative humidity approaching the deliquescence relative humidity point, with thickness of 3 to 5

monolayers for 50 nm particles (Hsiao et al., 2016). By incorporating these various conditions, species, and reactions, we gain

further understanding of the effects of SOA on particle growth.


## 2.1 Volatility of organic species (as Heading 2)

Products found in SOA are regularly quantified based on their volatility, which is expressed in terms of saturation concentration

($C^*$) in $\mu g\ m^{-3}$. A study by Donahue et. al. (2012) shows that ambient biogenic emissions contain many species which are

highly volatile ($C^* = 10^6\ \mu g\ m^{-3}$). Although these components are too volatile to partition/condense onto existing particles,

oxidation reactions in the atmosphere can produce lower volatility products from these reactants (Chen et al., 2011; Xavier et

al., 2019). For the ozonolysis of α-pinene reaction specifically, SOA products have been shown to range in volatility ($C^*$) from

$< 10^{-1}$ to $> 10^6\ \mu g\ m^{-3}$ (Donahue et al., 2012). For this model, volatility bins are simplified into three classes as follows: NVOCs,

which are non-volatile organics which have a $C^*$ of $10^{-4}\ \mu g\ m^{-3}$ and can condensationally grow particles, SVOCs, which are





semi-volatile organics whose volatilities are $10^0 \leq C^* \leq 10^3 \, \mu g \, m^{-3}$; and OVOCs, which are the remaining oxidation products
having a $C^* > 10^3 \, \mu g \, m^{-3}$ and are too volatile to grow particles. The current model incorporates one NVOC, one OVOC, and
four SVOC species, each with a specified volatility. These volatilities are listed in Table 1 alongside their corresponding
product yields. Molar yields of non-volatile ozonolysis products have been measured between 3.5 and 7 % through $NO_3^-$ Cl-
APi-TOF gas phase measurements in a high volume chamber (Ehn et al., 2014; Sarnela et al., 2018). The product yield (fraction
of α-pinene molecules that react with ozone to give a product molecule in the indicated volatility bin) for NVOC ($C^* < 10^{-3}$
$\mu g \, m^{-3}$) is set to 5 % based on these findings. Product yields for SVOCs and OVOCs are then derived through a study by
Trump and Donahue (2014), where volatility-based yields ($10^0 \leq C^* \leq 10^3 \, \mu g \, m^{-3}$) were fit to SOA aerosol mass yields utilizing
an equilibrium model.

| "X" VOC Designation | NVOC | SVOC$_0$ | SVOC$_1$ | SVOC$_2$ | SVOC$_3$ | OVOC |
|---|---|---|---|---|---|---|
| Volatility ($C^*$; $\mu g \, m^{-3}$) | $10^{-3}$ | $10^0$ | $10^1$ | $10^2$ | $10^3$ | $>10^3$ |
| Molecular Product Yield (%) | 5.0 | 4.0 | 7.0 | 9.0 | 15.0 | 60.0 |
| Product Mixing Ratios (molecules cm$^{-3}$) | $1.0 \times 10^7$ | $8.0 \times 10^6$ | $1.4 \times 10^7$ | $1.8 \times 10^7$ | $3.0 \times 10^7$ | $1.2 \times 10^8$ |

**Table 1. Product distribution for α-pinene ozonolysis used to model particle growth.**


## 2.2 Product mixing ratios (as Heading 2)

For modelling single particle growth under atmospherically relevant conditions, constant values for the product mixing ratios
are used as shown in Table 1. These values are consistent with what might be observed in a boreal forest during new particle
formation (Vestenius et al., 2014). For modelling growth under typical flow tube conditions, product mixing ratios are time
dependent and calculated according to Eq. (1):

$$[XVOC]_{g,t+\Delta t} = [XVOC]_{g,t} + k_{II}[\alpha P]_t[O_3]_t y_{XVOC}\Delta t - k_{WL}[XVOC]_{g,t}\Delta t - k_{CS}[XVOC]_{g,t}\Delta t \qquad (1)$$

where $[XVOC]_{g,t}$, $[\alpha P]_t$, and $[O_3]_t$ are the respective mixing ratios at time t, $\Delta t$ is the time increment, and $y$ is the molar yield
of the respective VOC for α-pinene ozonolysis. Here, three processes are represented; the oxidation of α-pinene by ozone
based on a second order rate constant ($k_{II}$), the loss of products to the inner walls of the flow tube ($k_{WL}$), and loss of products
to the condensation sink ($k_{CS}$). The [XVOC] designation represents each of the six species in Table 1: NVOC, SVOC$_i$, and
OVOC, i.e. a separate equation for each product volatility. The development and application of this equation for flow tube
modelling has been described elsewhere in detail (Krasnomowitz et al., 2019).



## 2.3 Modelling particle growth (as Heading 2)

The amount of seed particle growth obtained for a given simulation is evaluated with respect to the partitioning/condensation of organic species onto the particle and/or into the particle phase (where applicable). Modelling calculations are performed reclusively, updating gas- and particle-phase concentrations every tenth of a second over the timescale of the simulation. Particle-phase concentrations for each species are calculated according to Eqs. (2-4) at each timepoint based on the gas-phase product yields discussed previously.

$$[NVOC]_{P,t+\Delta t} = [NVOC]_{P,t} + \frac{c}{2}\gamma[NVOC]_{g,t}\frac{S_P}{V_P}\Delta t - k_D[NVOC]_{P,t}[SVOC]_{P,t}\Delta t \tag{2}$$

$$[SVOC]_{P,t+\Delta t} = [SVOC]_{P,t} + \frac{c}{2}\gamma[SVOC]_{g,t}\frac{S_P}{V_P}\Delta t - k_D[NVOC]_{P,t}[SVOC]_{P,t}\Delta t \tag{3}$$

$$[DIMER]_{P,t+\Delta t} = [DIMER]_{P,t} + k_D[NVOC]_{P,t}[SVOC]_{P,t}\Delta t \tag{4}$$

Here, $[XVOC]_{P,t}$/$[DIMER]_{P,t}$ are the respective particle-phase concentrations at time t, $[XVOC]_{g,t}$/$[DIMER]_{g,t}$ are the respective gas-phase concentrations at time t, c is the mean thermal velocity, $\gamma$ is the uptake coefficient (C* dependent), $S_p$ is the surface

area of the particle, $V_p$ is the molecular volume of the particle, $\Delta t$ is the time increment, and $k_D$ is the second order rate constant for dimer formation. Dimerization rate constants have been reported on the order of $10^{-4}$ to $10^{-2}$ $M^{-1}s^{-1}$ for various reactions of hydroperoxides and aldehydes (Ziemann and Atkinson, 2012). Of the many products formed during SOA formation, approximately 50 % of the mass formed from α-pinene has been reported to have a peroxide functionality (Docherty et al., 2005). The rate constant for all simulations in this study is held at $10^{-2}$ $M^{-1}s^{-1}$. The saturation ratio ($S_d$) determines how well a

gas-phase compound partitions into the particle phase and is defined as the ratio of gas-phase mixing ratio to the saturation mixing ratio. A high ratio ($S_d \gg 1$) is found in species that condensationally grow particles, such as NVOCs. As SVOCs partition between the gas and particle phase, they grow particles at a slower rate due to a $S_d \ll 1$. It is important to note that Eq. (3) is written in terms of this case, which is applicable for all simulations in this study. However, if and when $S_d \geq 1$ for SVOC, no additional flow into the particle phase would occur unless the formation of DIMER shifted the equilibrium by

depleting the particle-phase concentration. Particle-phase reactions responsible for dimer formation are simplified and represented by the term $k_D[NVOC]_{P,t}[SVOC]_{P,t}\Delta t$ in these equations. However, this term includes any combination of NVOCs and SVOCs able to form a DIMER in each volatility bin. Each organic molecule and/or compound with a low enough volatility to remain in the particle phase is summed at each timepoint, representing any increase in particle-phase volume ($V_P$) over the previous timepoint. This is represented by Eq. (5):

$$V_{P,t+\Delta t} = V_{P,t} + [NVOC]_P V_{NVOC}\Delta t + [SVOC]_P V_{SVOC}\Delta t + [DIMER]_P V_{DIMER}\Delta t \tag{5}$$

where $V_{NVOC}$, $V_{SVOC}$, and $V_{DIMER}$ are the respective volumes attributing to particle growth. For effloresced seeded simulations, the initial $V_P$ is equal to zero as the seed is considered to have a solid core, increasing as organic matter condenses and forms an organic layer. In the case of water monolayers being added to the same seed particle, this value is equivalent to the volume of water on the surface of the particle. For deliquesced seed particles, $V_p$ is equal to the entire initial volume of the particle at

the beginning of the simulation. Diameter growth is of the particle is determined as shown by Eq. (6):





$$d_{t+\Delta t} = 2 \left( \sqrt[3]{\frac{3(V_{P,t+\Delta t})}{4\pi}} \right) \qquad (6)$$

**2.4 Growth yield (as Heading 3)**

Growth yield (GY) is defined as the fraction of all oxidation products colliding with the particle surface that are actually taken up into the particle causing it to grow, as defined by Eq. (7):

$$GY_t = \left( \frac{NVOC+SVOC_i+OVOC \text{ taken up into the particle} \text{OC uptake collisions}}{NVOC+SVOC_i+OVOC \text{ striking the particle surface} \text{Total surface collisions}} \right)_t \qquad (7)$$

GY is calculated for each time increment based on the amount of growth caused by each oxidation product during that increment. GY includes the full NVOC yield (except for particles $\ll$ 20 nm, where the Kelvin effect causes some NVOC molecules to remain in the gas phase) plus a portion of the $SVOC_i$ yield depending upon how significant partitioning is and how much of the partitioned $SVOC_i$ undergoes reaction in the particle phase to produce a non-volatile DIMER. In this study,
OVOC does not contribute to GY since its concentration in the particle phase is too small to efficiently form DIMER.

**3 Single particle growth under atmospherically relevant conditions (as Heading 1)**

**3.1 Particle growth with and without particle-phase chemistry (as Heading 3)**

The first series of calculations examines the effects of condensation, partitioning, and oligomerization reactions on particle growth under atmospherically relevant conditions, which serves as a base case for comparison to flow tube simulations. All
simulations in this section utilize the product mixing ratios listed in Table 1 and are held constant throughout the growth of the particle. Figure 1a shows a 5 nm effloresced ammonium sulphate seed particle grown to 100 nm by condensation and partitioning alone. Here, GY remains constant at 5 % throughout the simulation as NVOC condensationally grows the particle. While SVOC partitions between the gas and particle phase, it is too volatile to have any significant contribution to particle growth. Figure 1b shows the same simulation plus the potential for DIMER formation. Throughout this study, DIMER
formation is noted as $D.SV_0$, where a DIMER can only be formed between two SVOC molecules in the particle phase with a $C^*$ equal to $10^0$ µg m$^{-3}$. Note that the GY curve starts at 5 %, the same as if condensation were the only process attributing to growth. Since SVOC partitioning requires a particle phase, the absence of SVOCs lead to no immediate effect on particle growth due to particle-phase chemistry. However, once approximately one monolayer of NVOCs builds up on the particle's surface, SVOCs can begin partitioning and DIMER formation follows quickly afterwards. This speeds up particle growth (note
the shorter time scale to 100 nm) and results in an increase in GY with increasing particle size.

Growth rate (GR) is a measure of how quickly a particle grows. Since this term is determined by the size of the entire particle, it is subject to particle losses to the reactor walls and condensation sink, as well as the effect of molecular diffusion on vapor molecule transport to the particle's surface (mass flux correction factor, $\beta_d$, becomes less than 1). Growth yield allows for GR





to be represented independent of these factors, which affect the GR in a flow tube independently of the actual chemical

processes that contribute to growth.

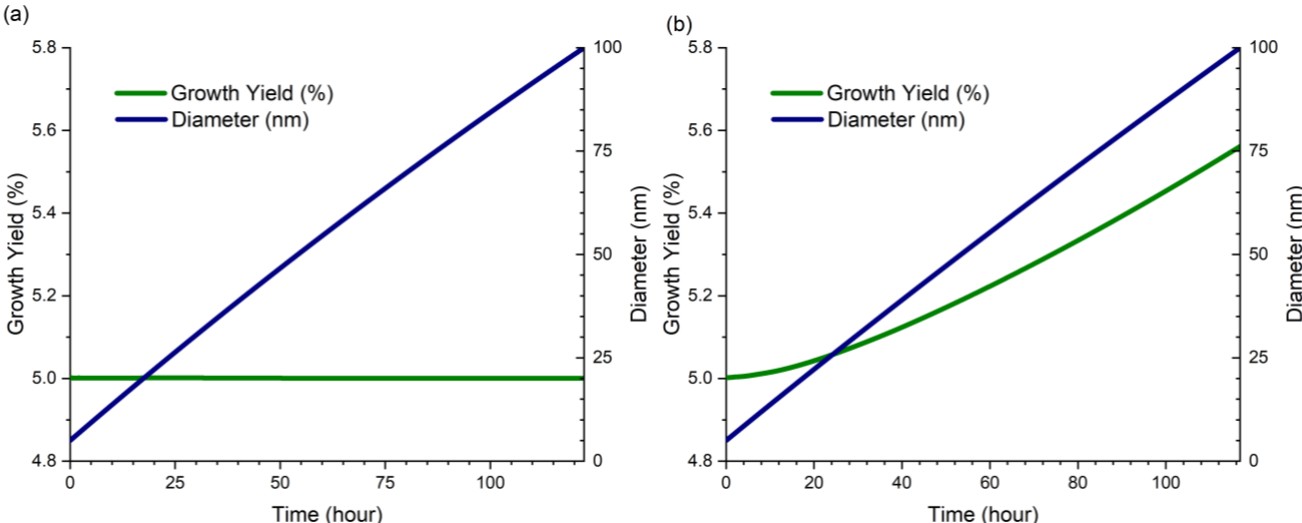

**Figure 1: Growth yield and diameter vs. time for single effloresced ammonium sulphate seed particle with an initial diameter of 5 nm. (a) Condensation of NVOCs and partitioning of SVOCs only. (b) SVOC DIMER formation in addition to condensation and partitioning.**

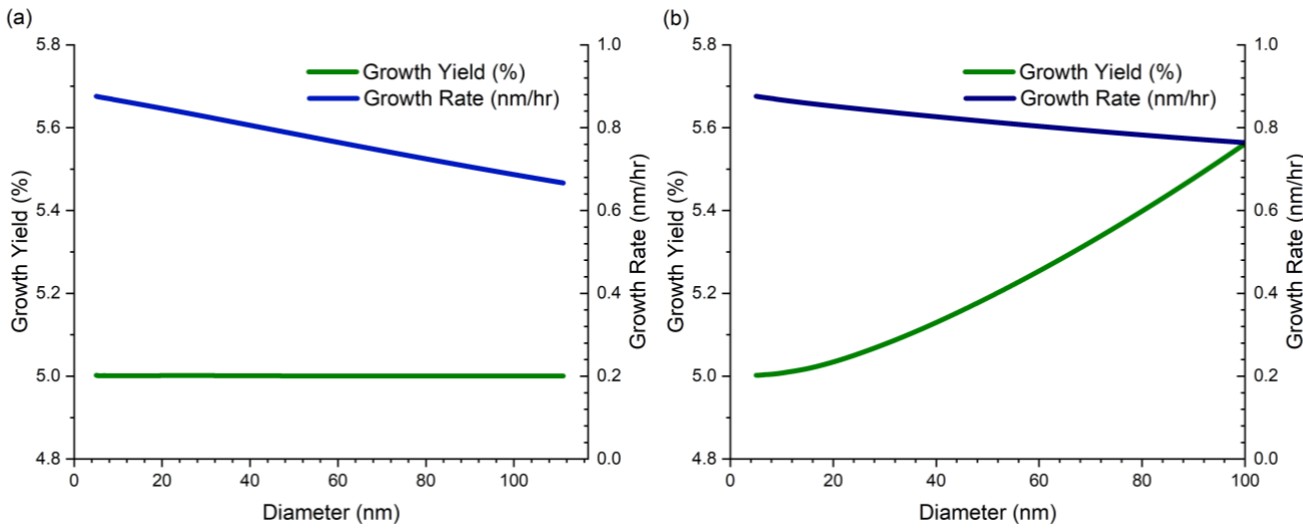


**Figure 2: Growth yield and growth rate vs. time for a single effloresced ammonium sulphate seed particle with an initial diameter of 5 nm. (a) Condensation of NVOCs and partitioning of SVOCs only. (b) SVOC DIMER formation in addition to condensation and partitioning.**





Figure 2 shows the GY re-represented and GR for the same two simulations in Fig. 1. Figure 2a shows a steady decrease in

GR with increasing particle size owing to the decrease in mass flux of NVOC and SVOC molecules to the particle surface. In

Fig. 2b, the decrease in GR is less substantial due to the increase in GR due to DIMER formation increasing with particle size.

GY is independent of both GR curves and allows for a clear understanding of how NVOCs, SVOCs, and DIMER formation

attribute to particle growth.

**3.2 Growth as a function of seed particle size and phase (as Heading 3)**

The next series of calculations explores the dependence of GY on both the starting seed particle size and phase (effloresced or

deliquesced). Figure 3 shows the growth yield with respect to particle size for 5, 20, 40, 60, and 80 nm ammonium sulphate

seed particles either effloresced (a) or deliquesced (b). All simulations in Fig. 3 involve the condensation of NVOCs,

partitioning of SVOCs, and DIMER formation (D.SV$_0$). For simulations of condensation and partitioning alone across all seed

sizes   and   phases,   GY   remained   constant   at   5   %   and   therefore   are   not   included   in   this   figure.

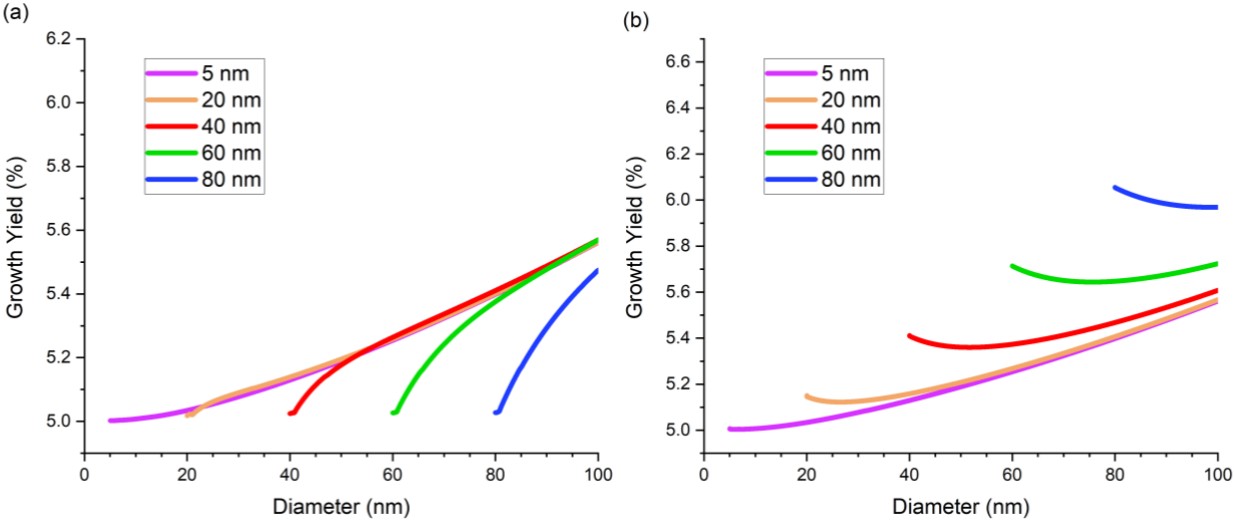

**Figure 3: Growth yield for a single (a) effloresced and (b) deliquesced ammonium sulphate seed particle growing from its initial size 100 nm. Condensation, partitioning, and DIMER formation are all included.**

Again, the GY for 5 nm seed particles are re-represented here from Fig. 1 for the respective seed phase. As discussed

previously, a slight period of growth at the beginning of each effloresced seed particle GY curve in Fig. 3a remains constant

at 5 % due to the immediate uptake of NVOCs but absence of a sufficient enough organic phase on the particle to allow for

SVOC uptake and subsequent DIMER formation. This period lasts for each seed particle until the diameter increases by

approximately 1.62 nm. This is consistent with an organic monolayer of NVOCs calculated to have a thickness of 0.81 nm on





a seed particle based on a molar mass of 200 g mol$^{-1}$ and a density of 1.2 g mL$^{-1}$. Once each effloresced seed particle has an

organic layer capable of supporting particle-phase chemistry, SVOC uptake begins and is followed almost immediately by

DIMER formation, attributing to the increase in GY. For each seed size, the growth yield then converges toward the maximum

yield attainable under these conditions. Deliquesced seed particles, shown in Fig. 3b, can take up both non-volatile and semi-

volatile products immediately due to the available particle-phase volume being the entire particle rather than the organic layer

of an effloresced seed. Rapid SVOC partitioning and DIMER formation at the onset of growth give a high starting growth

yield which increases with seed size as the increase in surface area and volume allow for greater SVOC uptake. As additional

SVOC molecules partition into the particle phase and react, the particle grows and the growth yield converges toward the

effloresced particle growth yield.

Particle-phase chemistry can accelerate particle growth by increasing the amount of non-volatile organic material in the particle

phase, which can be represented by a time dependent change in growth yield. When condensation is the only process growing

a particle, GY remains unaffected by seed particle size and phase. However, GY is strongly affected by seed particle size and

phase when particle-phase chemistry is present. Reaction time is also an important parameter since Fig. 3 shows that GY

changes as the particle grows. With this fundamental understanding of particle-phase chemistry and GY, this model can be

applied to simulations of an experimental apparatus.

**4 Particle growth in a flow tube reactor (as Heading 1)**

In this section, particle growth is modelled under conditions typically used in our flow tube reactor, whose design and

performance are described in detail elsewhere (Krasnomowitz et al., 2019). Briefly, size selected ammonium sulphate seed

particles (solid or liquid-like; 40, 60, or 80 nm dia.) are introduced in the flow tube along with gas-phase BVOC (α-pinene in

our initial experiments), ozone, cyclohexane (hydroxyl radical scavenger), and water vapor (relative humidity control). Particle

residence time in the flow tube is approximately four minutes. Experimental conditions are chosen such that particles exiting

the flow tube reactor have increased from their initial diameters by about 1 to 8 nm, which can be measured with high precision

using a Scanning Mobility Particle Sizer (TSI, Inc.). This range of diameter increase corresponds to a growth rate between

about 15 and 120 nm hr$^{-1}$. For comparison, ambient particle growth rates (and the simulations in the previous section, which

were based on ambient conditions) are on the order of 1 to 10 nm hr$^{-1}$ for new particle formation events. To achieve the desired

amount of particle growth, we typically perform flow tube experiments with an α-pinene mixing ratio of 11 ppbv and ozone

mixing ratios between 30 to 300 ppbv. Similar parameters have been used by others (Pathak et al., 2007) as well as us

(Krasnomowitz et al., 2019) to experimentally study SOA formation by α-pinene ozonolysis.

**4.1 Particle growth in a flow tube with and without particle-phase chemistry (as Heading 3)**

Simulations within this section apply the growth model to specific experimental conditions found within the flow tube reactor.

A single ammonium sulphate seed particle is simulated traveling through the reactor, and again the partitioning and



condensation of NVOCs and SVOCs are active in all cases, with the ability to add in particle-phase DIMER formation reactions between two SVOC molecules with a C* equal to $10^0$ μg m$^{-3}$ where noted. Notable changes to the parameters of the model include reducing the timescale to four minutes and utilizing time dependent product mixing ratios calculated through Eq. (1)

based on the molecular product yields in Table 1. For the data discussed in Figs. 4 and 5, an ozone mixing ratio of 200 ppbv was used in calculating all product mixing ratios. The change in growth yield and diameter (Δd) for 40 nm effloresced seed particle sent through the flow tube reactor under four different conditions are shown in Fig. 4. The condensation of NVOCs and partitioning of SVOCs are the only processes active in Fig. 4a. Here, GY quickly increases and remains constant at 5 % throughout the timescale of the simulation, consistent with condensational growth. Fig. 4b shows the addition of particle-phase

chemistry (D.SV$_0$) to the same seed particle conditions. Note that the GY curve begins similar to that of Fig. 4a, as only NVOCs can condense onto the particle in the absence of a reactive particle phase. However, once the volume of the organic coating on the particle surface is sufficient, DIMER formation begins and quickly increases the GY. This is attributed to the rate of DIMER formation, which is fast enough that essentially all the SVOC$_0$ molecules taken up into the particle's organic layer react to form DIMERs. The GY continually increases until reaching approximately 9 %, at which point it remains constant

for the remainder of the simulation. In this case, the molecular product yield of 4 % for SVOC$_0$ is added to the existing NVOC molecular product yield of 5 % to give the total GY. Once the uptake of SVOC$_0$ is maximized, any further uptake and DIMER formation is limited by the collision rate. By the end of the simulation, GY is approximately 9.1 to 9.2 % owing to the small number of partitioned SVOC molecules that contribute minimally to particle growth.

The first two simulations utilize an effloresced seed particle, which has a solid phase core, preventing SVOC uptake until an organic layer of NVOCs builds up on the particle's surface sufficient to allow for SVOC partitioning to begin. However, studies have shown that approximately 3 to 5 monolayers of water molecules can remain on effloresced seed particles of 50 nm under higher relative humidity conditions, but still below that of deliquescence for ammonium sulphate (Hsiao et al., 2016). The presence of water allows for immediate uptake of SVOCs into the particle phase. For the simulation shown here in Fig.

4c, four monolayers of water are added to the surface of the starting effloresced seed particle and grown over the time scale of the flow tube reactor. The thickness of the water layer in the simulation is 1.54 nm based on a density of 1.00 g mL$^{-1}$ and a molecular diameter of 0.385 nm. Note that the change in diameter has been adjusted so that the initial diameter includes this water layer in addition to the dry seed. Here, a clear distinction can be made between the presence and absence (Fig. 4b) of a liquid-like coating at the beginning of particle growth. A significant spike occurs at the beginning of the simulation due to the

rapid formation of DIMERs from SVOCs that are immediately taken into the particle phase. Once the partitioning of SVOCs equilibrates, the GY returns to approximately 5.5 % in each case at the approximate time point of 10 s. As seen in Fig. 4b, GY reached this point at a much slower rate, approximately 90 seconds into the simulation, for the same particle-phase reaction conditions. Additionally, the maximum uptake of SVOC$_0$ (GY becomes 9 %) is reached at an earlier point on the time scale. These results show how even a minimal volume of particle phase (compared to the entire effloresced seed particle) can

significantly impact how particle-phase chemistry grows a particle.





**Figure 4: Growth yield and diameter change for a single 40 nm dia. ammonium sulphate seed particle as it travels through the flow tube, growing by (a) condensation and partitioning alone, and (b-d) with DIMER formation included.**
**Plots shown in (a) and (b) are for an effloresced particle, (c) an effloresced particle with four surface monolayers of water present on the surface, and (d) a deliquesced particle.**



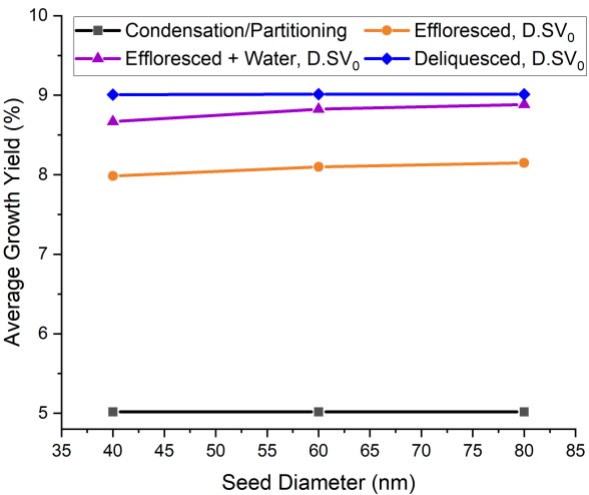

**Figure 5: Average growth yield (GY) vs. initial seed particle diameter. GY is averaged across the timescale of the flow**
**tube with the GY at each time point weighted by the diameter growth rate at that time point. Lines are drawn as an**
**aid to the eye. Gray – deliquesced particle growing by condensation and partitioning alone. Orange – deliquesced**
**particle growth by DIMER formation as well. Purple – deliquesced particle with four surface monolayers of water**
**growing by condensation, partitioning and DIMER formation. Blue – effloresced particle growing by condensation,**
**partitioning, and DIMER formation.**

Deliquesced seed particles are considered to be fully in the liquid phase and therefore are modelled where the entire volume
of the particle is available for particle-phase chemistry from the start of the simulation, as shown in Fig. 4d. Here, the same
conditions for condensation, partitioning, and particle-phase chemistry are present as discussed in Figs. 4b and 4c. Note the
similar trend in GY to that of Fig. 4c, where an instant increase in GY is followed by an equilibration period until $SVOC_0$
reaches its maximum uptake. This maximum is now reached on the order of 20 seconds, approximately eight times faster than
a solid core effloresced seed and six times faster than an effloresced seed with 4 monolayers of water. With $SVOC_0$ able to
contribute to particle growth for nearly the entire timescale of the flow tube, particle growth is substantially increased from
that of condensation alone, with an increase in change in diameter from approximately 2.9 to 5.0 nm.

The size dependence of these conditions was also examined for 60 and 80 nm seed particles. Since the overall trends are very
similar, the results of the simulations are summarized in Fig. 5, where the average growth yields are taken across the entire
time scale of four minutes. Throughout this study, the average GY was determined by averaging the GY at each time point
weighted by the diameter growth rate at the same point, giving larger weight to GY values that contribute the most to diameter
growth. As would be expected, the GY remains constant at 5 % for all seed particle sizes when growth occurs by condensation



and partitioning alone. For simulations that include growth by particle-phase chemistry, GY increases as the amount of liquid-like volume increases. GY is lowest for effloresced particles without any surface water, and highest for deliquesced particles where the entire volume is accessible. Because particle growth in much of the flow tube is at the maximum rate for the conditions of these simulations (GY of 9 % in Fig. 4), it is not surprising that the particle size dependence of average GY in Fig. 5 is relatively flat for the cases where particle-phase chemistry occurs. Careful inspection of these plots shows that there

is a slight increase in GY with increasing particle size, as would be expected based upon an increasing volume-to-surface area ratio with increasing particle size.

It is instructive to compare the growth yields and diameter changes for atmospheric conditions in Fig. 1 to the flow tube environment in Fig. 4. First, if only condensational growth due to NVOC is involved, the two conditions give the same result

i.e. both give the same GY (5 % for these simulations), and this value remains constant as time and particle diameter increase. However, when DIMER formation inside the particle occurs, the two conditions can give very different results. The higher precursor mixing ratio conditions of the flow tube cause the GY to increase quickly with time and particle diameter, and particle growth changes from volume-limited kinetics (GY increases with increasing particle diameter) to surface-limited kinetics (GY reaches its maximum value, 9 % for these simulations) and then remains constant.


### 4.2 Growth in a flow tube as a function of ozone mixing ratio (as Heading 3)

Thus far, flow tube simulations have focused on particle-phase chemistry for a constant ozone mixing ratio of 200 ppbv. Simulations in this section expand the range of mixing ratios to range between 50 and 300 ppbv, which have been used previously to investigate condensational growth of effloresced seed particles (Krasnomowitz et al., 2019). Figure 6 shows the

growth yields as a function of time inside the flow reactor for 40 nm dia. effloresced (Fig. 6a) and deliquesced (Fig. 6b) seed particles exposed to various ozone mixing ratios. For reference, the 200 ppbv plots in Figs. 4b and 4d are re-represented in Figs. 6a and 6b, respectively. In both cases, decreasing the ozone mixing ratio elongates the shape of the time plot. For the deliquesced particles in Fig. 6a, the maximum 9 % GY is no longer reached for the lowest two mixing ratios. For the effloresced particles in Fig. 6b, the maximum GY is always reached, but later and later in the flow tube as the ozone mixing ratio decreases.

The simulations in Fig. 6 are summarized in Fig. 7 by taking the average GY across the time scale of the simulation for each ozone mixing ratio. Results for two additional simulations are also included in Fig. 7: effloresced seed particles growing by condensation and partitioning alone (e.g. the simulation in Fig. 4a for 200 ppbv ozone) and effloresced seed particles containing four water monolayers on the surface that are able to grow by particle-phase reaction (e.g. Fig. 4c for 200 ppbv ozone). The average GY remains constant at 5 % for particles growing by condensation and partitioning alone at all ozone mixing ratios.

In contrast, GY increases with increasing ozone mixing ratio for the other three simulations where particles are able to grow by particle-phase chemistry. GY also increases with increasing particle volume available for particle-phase chemistry.





Deliquesced seed particles show the highest GY for all ozone mixing ratios since particle-phase chemistry can occur throughout the entire particle volume.

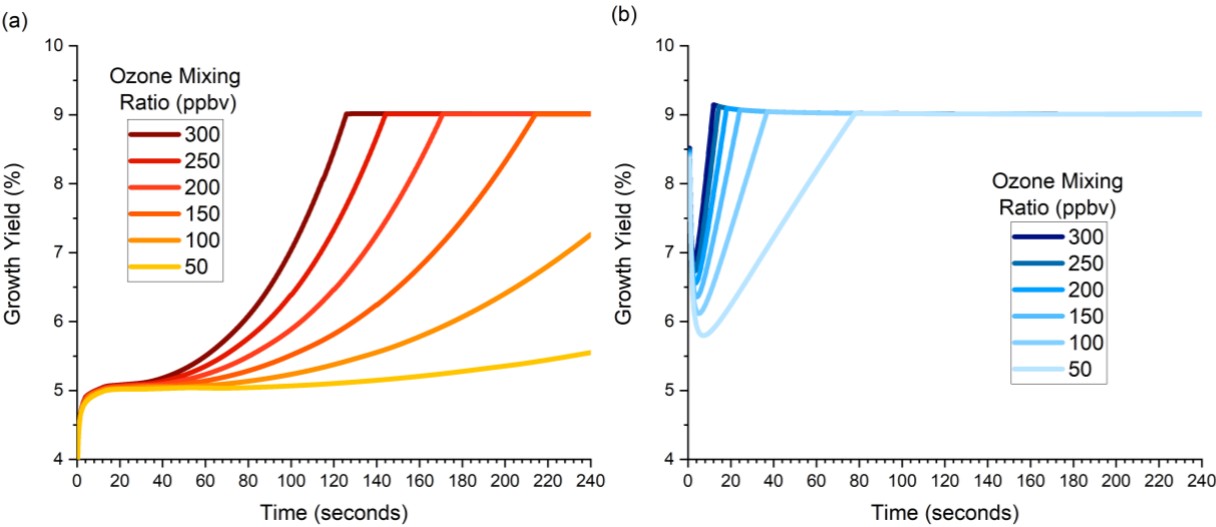

**Figure 6: Growth yield vs. time in the flow tube for a single 40 nm ammonium sulphate seed particle: (a) effloresced, and (b) deliquesced. Condensation, partitioning, and DIMER formation are all included. Ozone mixing ratio is shown with increasing colour intensity corresponding to increasing mixing ratio.**

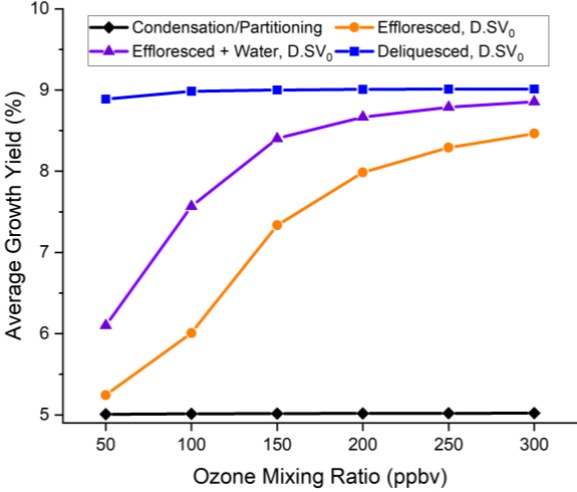

**Figure 7: Average growth yield vs. ozone mixing ratio for the 40 nm dia. ammonium sulphate seed particles and conditions shown in Figures 4 and 5. Lines are drawn as an aid to the eye.**





While the ability of particle-phase chemistry to increase GY generally increases with increasing particle diameter, Fig. 5 shows
that this effect can be muted if the reaction is very fast and growth becomes surface- rather than volume-limited over a
significant portion of the flow tube. In these situations, decreasing the ozone mixing ratio can enhance the particle size
dependence by lowering the reaction rate and moving away from surface-limited kinetics. This is illustrated in Fig. 8 for
deliquesced particles from 40 to 80 nm, which represents a "worst-case" situation for the various simulations studied. When
the ozone mixing ratio is high, the GY for all three particle sizes converges toward the maximum value (9 %). When the mixing
ratio is lowered, the GY for the three sizes begin to diverge because volume-limited kinetics encompass a greater portion of
time in the flow tube. Even when the diameter dependence of the GY in a flow tube obscures volume-limited kinetics (Fig. 5),
the existence of particle phase chemistry can be inferred from the GY dependence on seed particle composition and/or phase
e.g. average GY of deliquesced vs. effloresced particles in Fig. 5.

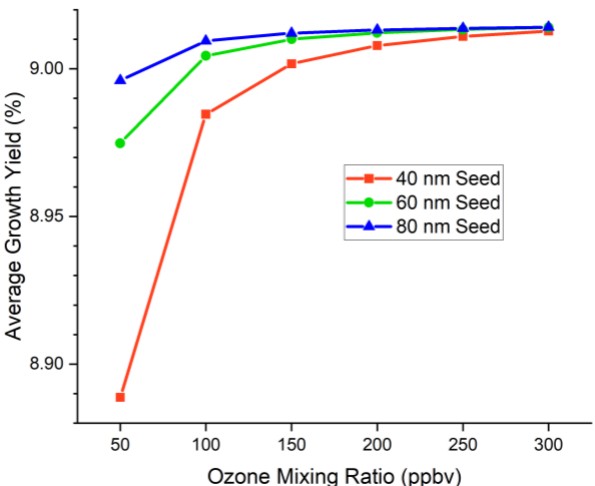

**Figure 8: Average growth yield vs. ozone mixing ratio for 40, 60, and 80 nm deliquesced ammonium sulphate seed**
**particles. Condensation, partitioning, and DIMER formation are all included. Lines are drawn as an aid to the eye.**

### 4.3 Impact of interfacial water (as Heading 3)

As shown in Fig. 7, the addition of water monolayers on the surface of an effloresced seed particle enhances DIMER formation
significantly. An in-depth examination of this effect is shown in Fig. 9, where the growth due to condensation, partitioning,
and particle-phase reactions of 40 nm seed particles of varying phase and surface water content are simulated. Here, effloresced
seed particles take the longest amount of time to reach complete uptake of NVOCs and SVOCs for particle growth. Monolayers
of water ranging from one to five layers thick are added to the same particle, significantly increasing the uptake of SVOCs





over that of the effloresced seeded simulation. The simulation of a deliquesced seed particle is included for comparison to a

particle of the same size where the entire volume is considered to be the particle phase. The plots in Figs. 5, 7, and 9 illustrate

the relatively large impact of surface water on DIMER formation and highlight the importance of fully characterizing the air-

particle interface in both laboratory and field settings.

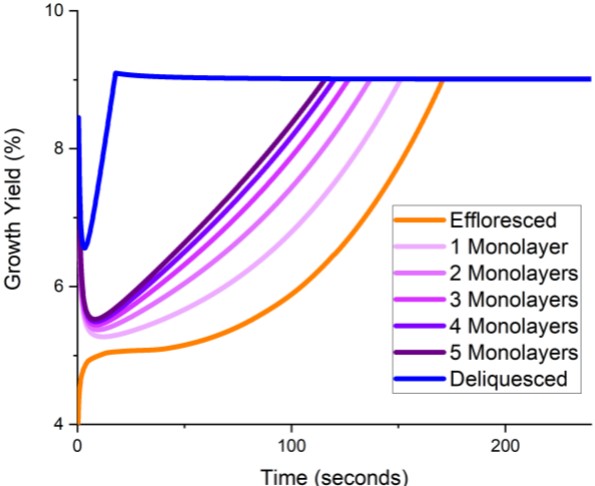

**Figure 9: Growth yield vs. time in the flow tube for effloresced seed particles containing 1-5 water monolayers on the surface, and for a deliquesced seed particle. Condensation, partitioning, and DIMER formation are all included.**


### 5 Interpreting Flow Tube Measurements (as Heading 1)

In the simulations so far, the diameter growth of particles and corresponding GY were calculated as a function of time using a

specific chemical model of the growth processes. In a flow tube experiment where growth is compared for various gas- and

particle-phase conditions, the opposite occurs. Diameter growth is measured at the outlet of the flow tube, which corresponds

to a single time-point in the process, and from that an average GY is determined and interpreted in the context of various

chemical growth models. Because no specific model is assumed, the various oxidation products represented by Eq. (1) are

replaced by a single growth species:

$$[COV]_{g,t+\Delta t} = [COV]_{g,t} + k_{II}[\alpha P]_t[O_3]_t y_{COV}\Delta t - k_{WL}[COV]_{g,t}\Delta t - k_{CS}[COV]_{g,t}\Delta t \qquad (8)$$

where $y_{COV}$ is the yield of condensable organic vapor (COV) that is able to grow the particles. The modelling process fits the

lone adjustable parameter ($y_{COV}$, which is also defined as the growth yield, GY) so that the calculated diameter change at the

exit of the flow tube matches the measured value. More details on modelling experimental data can be found elsewhere

(Krasnomowitz et al., 2019). Similar to the model in Sect. 2, COV in Eq. (8) encompasses all NVOC plus a portion of SVOC





that is able to enter the particle and stay there. Unlike the model in Sect. 2, $y_{COV}$ is assumed to be constant across the length of the flow tube since only a single growth species is assumed. The first question to ask is how well the GY calculated from the

approach in Eq. (8) matches the average GY in Figs. 5, 7 and 8. To answer this question, we used the modelling approach of Sect. 2 to determine the diameter change at the time-point particles would exit the flow tube, and then we determined the COV by the approach in Eq. (8). Figure 10 compares the average GY to the COV yield for 40, 60, and 80 nm seed particles under the condition of 200 ppbv ozone. When DIMER formation is excluded from the simulation, both the average GY and COV yield give the same value – 5 % which corresponds to the NVOC yield. When DIMER formation is included in the simulation

and the conditions are such that surface-limited kinetics exist throughout the flow tube, both the average GY and COV yield give the same value: 9 % which corresponds to the sum of NVOC and SVOC yields. In between these two extremes where volume-limited kinetics exist in at least part of the flow tube, the average GY and COV yield track each other closely and deviate at most by at most a few percent of the respective values. At a yield of 5 %, all values are equivalent as the same condensational growth process is being modelled in both methods. Once particle-phase chemistry is involved in the growth

process, the average GY deviates from a 1:1 ratio with COV yield, with the largest difference being for effloresced seed particle growth (3.2 to 3.4 % across seed sizes). In comparison to other particle-phase simulations, effloresced seed particles are grown by a lower yield of products for a longer period of time in the flow tube (e.g. the simulation in Fig. 4b), therefore requiring a lower COV yield to obtain an equivalent amount of growth. As the initial particle phase increases due to surface water or a liquid-like seed phase, the two methods converge towards a 1:1 ratio as growth is limited by the collision rate and particle-

phase reactions essentially contribute the same as condensation alone.

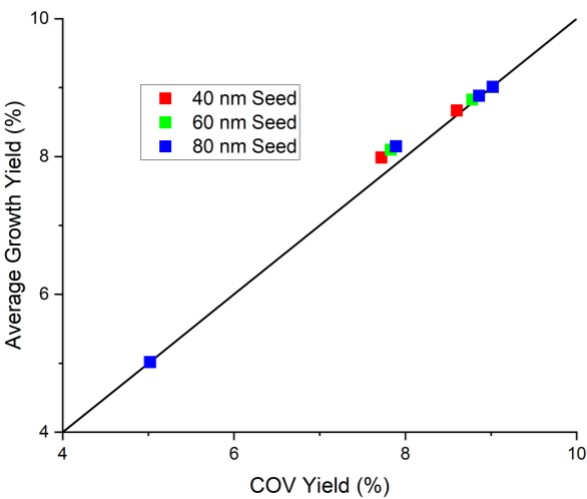

**Figure 10: Average growth yield from growth modelling vs. "experimentally determined" COV yield for 40, 60, and 80 nm seed particles under the various conditions of this study. All simulations shown in this figure have an ozone mixing ratio of 200 ppbv. The line shows a 1:1 ratio.**



A key aspect of flow tube experiments is that precursor mixing ratios are generally higher than ambient levels, which was discussed in some detail in Sect. 4.1 with regard to surface- vs. volume-limited kinetics. Elevated mixing ratios are not a problem for linear processes, in other words growth processes that are directly proportional to the concentration of the growth precursor, either in the gas phase or the particle phase. The example here is growth by condensation of NVOC (Figs. 1a, 2a, 4a, 5). In these cases, GY is constant over the residence time of the flow tube and is independent of precursor mixing ratio as

well as initial seed particle diameter and phase. Nonlinear processes, for example DIMER formation in this study which is proportional to the square of the SVOC concentration in the particle volume, are indicated by a precursor mixing ratio dependence of GY (Figs. 6 and 7). GY will also increase with increasing seed particle diameter as long as the reaction rate in the particle phase is volume-limited rather than surface-limited (Fig. 8). For nonlinear processes in the gas phase, for example $RO_2/RO_2^-$ chemistry in the gas phase, particle growth remains mass flux dependent, so GY is fully independent of seed particle

diameter, though it will still show a precursor mixing ratio dependence. This discussion highlights how both seed particle diameter and precursor concentration dependencies are needed to understand the kinetic limitations of particle growth in a flow tube experiment and to accurately extrapolate the expected growth back to atmospherically relevant conditions.

A final consideration is condensation sink. The simulations presented in this work have been based on a single particle growth

model. In practice, particle number concentrations in size-selected experiments are on the order of $10^4$ cm$^{-3}$ and the magnitude of the condensation sink is different for different size particles: 1.8 x $10^{-4}$, 2.5 x $10^{-4}$, and 3.1 x $10^{-4}$ s$^{-1}$ for 40, 60, and 80 nm seed sizes, respectively based on a number concentration of $10^4$ cm$^{-3}$. A benefit of the modelling approach adopted here is that GY remains essentially independent of condensation sink, whereas the diameter change can be condensation sink dependent. For example, the $\Delta d$ and average GY for an 80 nm seed particle without including the condensation sink are 4.04 nm and 7.06

%, respectively. By adding in the condensation sink for a particle concentration of $10^4$ cm$^{-3}$ in the same simulation, the $\Delta d$ is reduced to 3.93 nm and the growth yield is slightly reduced to 7.04 %.

## 6 Conclusions (as Heading 1)

Flow tubes provide an effective way to study growth as a function of seed particle size, composition, and phase state, as well

as other conditions such as precursor mixing ratios and relative humidity. This modelling study shows that through a combination of seed particle size and precursor mixing ratio dependencies, insight can be gained into kinetic limitations (volume- vs. surface-limited growth). These dependencies, in combination with other variables such as particle composition and phase state, and molecular composition measurements by mass spectrometry, provide additional constraints on the types of chemical processes responsible particle growth. Finally, the simulations show the disproportionate ability of one or a few

monolayers of water on the particle surface to enhance growth by providing a medium for particle-phase chemistry to occur, highlighting the need to characterize the air-particle interface.



**Author Contribution**

MT and DH performed the simulations for modelling particle growth. MT prepared the manuscript with contributions from all
authors.

**Competing Interests**

The authors declare that they have no conflict of interest.

**Acknowledgements**

This research was supported by two grants from the U.S. National Science Foundation, CHE-1904765 and AGS-1916819.



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
