# Peer review of "Modelling ultrafine particle growth based on flow tube reactor measurements"

_Atmospheric Measurement Techniques, 2022_

## Author Response (AR1)

**Detailed Author Responses to Reviewer Comments**

**Authors General Response**

We thank both reviewers for their comments, which gave us direction of how to better communicate the goals, results, and conclusions of this work in our revision. Before going into specific comments and changes, we focus on two key points that appear to be at the center of concern for both reviewers. Specific replies to individual reviewer comments appear after these general comments.

Key point #1: Modeling seems to be incomplete and/or oversimplified.

We see how confusion has arisen over this topic as there are two different types of "modeling" described in the paper, each with a very different intention that also affects how complete they are with respect to detailed models in the literature. First is the SOA formation "model" (condensation of nonvolatile compounds; oligomerization of semivolatile compounds that have partitioned into the particle phase). This modeling is meant to illustrate the complexity of what might be happening inside the flow tube, even though we are not able to directly measure it. Think of this model as simulated flow tube data that we wish to interpret. Second is the model referred to in the title of the manuscript – which is a generic model to interpret flow tube data when we don't know the detailed physicochemical processes and parameters needed to accurately calculate SOA formation. Hopefully, these concepts will become clearer in the remainder of this response.

The goal of this study is to investigate how best to represent complex particle growth kinetics within a flow tube reactor. In a typical flow tube experiment, one measures the input and output conditions to determine a time-averaged measure of growth, which may be difficult to interpret if the growth kinetics change as particles transit through the flow tube. In this work, we use a simplified single-particle growth model (referred to as a "simulation" in the revised manuscript – see below) for secondary organic aerosol (SOA) formation to illustrate how complex growth kinetics inside a flow tube can arise (Section 2). We then develop and assess a method to represent these complex growth kinetics when the details of SOA formation (chemical reactions on or inside the particle, molecular diffusion within the particle, etc.) are unknown (Section 5).

The SOA growth model used in this study is not meant to be a detailed chemical model of any specific system, though the model we use is not arbitrary but inspired by SOA formation from alpha-pinene ozonolysis. The SOA formation model is used to simulate complex growth kinetics. Accordingly, the essential features of this model are a surface-limited process (condensation of nonvolatile material), and a volume-limited process (in this case, dimer formation in the particle phase that transforms partitioned semivolatile compounds into nonvolatile dimer products), and a range of volume-limited reaction pathways and rates relative to the surface-limited process (multiple semivolatile gas-phase mixing ratios and volatilities that influence their concentrations in the particle phase and hence the dimer formation reaction rate). Of course, a detailed model for alpha-pinene SOA would include a broader range of ozonolysis products and particle phase reactions along with relevant physico-chemical properties, for example hindered molecular diffusivity in the particle phase – all of which would affect how

much growth occurs due to dimer formation.  However, such a detailed model is not needed to draw basic conclusions.

The "modeling" referred to in the title is the second type of modeling, which is used to interpret flow tube data as described in Section 5.  Here, we calculate a "growth factor" based on the diameter change of particles from inlet to outlet of the flow tube as simulated by the modeling procedure of Section 2.  This interpretation of flow tube data is based on deriving a single value for the growth factor from the change in particle diameter from inlet to outlet.  However, the SOA formation model (simulation) shows that even under simplified reaction conditions, the growth factor is NOT constant as particles move through the flow tube.  So the question arises (and is answered) how closely does the single number for growth factor in the interpretation modeling correspond to the actual range of growth factors in the flow tube?

Changes to the revised manuscript to incorporate these concepts:

General change #1: We avoid "model" as much as possible throughout the revised manuscript because of the above confusion.  In general, we refer to the SOA formation calculations as "simulations" and we refer to the analysis of flow tube "data" as a "method".

General change #2: We have changed the term "growth yield" to "growth factor" in the manuscript.  We hope that the change of wording will further emphasize that we're not trying to perform detailed modeling of a specific SOA reaction, but rather to define a parameter that empirically describes particle growth when no such detailed model exists.

Specific change #1: We have rewritten the abstract to include a succinct statement of the goal/significance our study at the very beginning: "*Flow tube reactors are often used to study aerosol kinetics.  The goal of this study is to investigate how best to represent complex growth kinetics of ultrafine particles within a flow tube reactor when the chemical processes causing particle growth are unknown*."

Specific change #2: We have completely rewritten paragraphs 4 and 5 of the introduction to expand upon this goal.  Excerpts from the fourth paragraph: "*The complexity of SOA chemistry as discussed above poses a challenge for studying particle growth with a flow tube reactor… Because of the complex time dependence of particle growth in a flow tube, simply reporting the growth rate (diameter increase per unit time) based on inlet-outlet size distributions and flow tube residence time is insufficient for predicting growth in other laboratory experiments and/or ambient air.  The goal of this study is to provide a framework for interpreting flow tube data that gives predictive capability*."  Excerpts from the fifth paragraph:  "*First, we simulate particle growth using a basic SOA formation model that incorporates both surface- and volume- limited growth pathways.  We use these simulations to illustrate complex growth kinetics that arise and how they are represented by the time dependence of GF… [Next,] we show how GF can be estimated directly from flow tube data (inlet-outlet size distribution, residence time, initial VOC and oxidant mixing ratios).  For the SOA simulations described above, we compare the estimated GF to the range of GFs actually inside the flow tube.  The estimation method is shown to be a robust way of representing complex growth kinetics from a flow tube experiment, without requiring prior knowledge of the specific chemical processes involved in SOA formation*."

Specific change #3: We have completely rewritten the first two paragraphs of Section 2. Excerpt from paragraph 1: "*Since we cannot directly measure particle size distributions at various locations inside the flow tube, particle growth must be simulated. The simulation described below contains four key elements: gas-phase kinetics (generation of molecular species capable of growing particles), a range of gas-phase mixing ratios (facilitates comparison of growth kinetics under atmospheric vs. flow-tube conditions), aerosol growth kinetics (uptake mechanisms of gas-phase molecules on/into the particle), and physicochemical processes and parameters typical of biogenic SOA formation.*" This paragraph goes on to justify the simulation approach including new literature references. Then, from paragraph 2: "*In order to make these simulations relevant to experimental investigations, we use molecular properties and processes typical of α-pinene SOA formation since this system is so well studied over the years. However, it should be understood that the simulations are not meant to accurately calculate the amount of α-pinene SOA formed.*"

Specific change #4: We have completely written Section 5 for context. From the first paragraph: "*Given this complexity, how does one extract useful growth information from an experiment when the processes leading to growth are poorly understood? In this section, we discuss a method to use flow tube data to determine GF without knowledge of the growth processes involved. For the simulations in Section 4, we compare GFs obtained from this interpretive method to the actual GFs from the simulations.*" From the final paragraph: "*It is important to realize that the simulations in Section 4 and the COV calculation in Section 5 are not simply "reverse" calculations of each other. The simulations in Section 4, though simplified relative to detailed SOA formation models for specific VOC precursors, incorporate numerous chemical details… None of these details are included in the COV calculations…*"

Specific change #5: We have written the equations in Section 5 as differential equations (rather than deltas) to distinguish them completely from those in Sections 2-4.

Specific change #6: Finally, we drive home these points at the end of the conclusions section: "*Since the specific reactions driving aerosol growth kinetics are often unknown or only partially understood for many SOA systems, an empirical calculation of COV growth factor based on outlet-inlet particle diameters in a flow tube experiment can give predictive capability for SOA growth. In the present study, empirical COV growth factors closely matched the actual GFs from SOA simulations, and one can expect that the difference between empirical and actual growth factors in flow tube experiments will be within typical experimental uncertainties.*"

We trust that these changes above will make the goal of the the study and the approach used to meet the goal easily understood by the reader.

Key point #2: Figures were consolidated to more effectively convey the key points and significance of this study.

As suggested by Reviewer 1, we limited the revision to five figures, which shortened the manuscript substantially. These five figures, described briefly below, were chosen to illustrate as

clearly and concisely as possible the key results and conclusions of the study.   While they are formatted differently from the original manuscript, the underlying calculations and results are the same as in the original manuscript.  The revised figures deal specifically with 40 nm dia. seed particles.  Figures, calculations, and discussion from the original manuscript dealing with other particle sizes were removed – we didn't think that the additional insight gained from different particle sizes justified the greater manuscript length, and they only served to dilute the major points we were trying to make.

Figure 1 – This figure consists of slight reformatting of Figures 1a and 1b in the original paper.  This figure introduces concept of growth factor in the context of particle growth under atmospheric conditions.  We included shaded regions in the revised figure to link growth factor with changing diameter growth rate as a prelude to studying particle growth in a flow tube – which involves a small slice of particle growth relative to an extended new particle formation event in the atmosphere.

Figure 2 – This figure consists of Figures 4a and 4b of the original paper, which simulates how particles are growing in the flow tube as opposed to atmospheric conditions.  Key points are: 1) one must use much higher mixing ratios than ambient conditions in order to obtain measurable growth over the short time period of the flow tube, and 2) growth factor changes greatly over the time period of the flow tube, illustrating the complexity of growth kinetics.

Figure 3 – This figure consists of Figure 6a in the original paper.  The idea here is that the high mixing ratios of reactants in the flow tube cause the limiting kinetics of particle growth (volume vs. surface-area limited) to be much different from that in the atmosphere.  Also, particle growth is nonlinear with increasing gas -phase mixing ratio, and surface-limited growth begins to win out over volume-limited growth.

Figure 4 – This figure consists of Figure 9 in the original paper.  It illustrates how aerosol liquid water can enhance particle growth, and in particular shows that even just a few monolayers of water on the surface of an effloresced particles have the ability to substantially increase the growth rate.

Figure 5 – This figure is similar to Figure 10 from the original paper.  However, the data points in the revised figure are for the specific simulations shown in revised Figures 2-4.  (The original figure showed data from different seed particle sizes, which are eliminated from the revision.)  This figure compares the COV growth factors obtained from interpretive method based on inlet-outlet change in particle diameter (Section 5) to the actual growth factors (time averages of the simulated GFs in Figures 2-4).  Interpretive modeling is shown to be a robust way of describing particle growth when the changing growth kinetics within the flow tube are unknown.  It can be used, for example, to empirically quantify the differences in particle growth due to changing precursor mixing ratios (Figure 3) or different amounts of aerosol liquid water (Figure 4).

We trust these revised figures give a more compact and effective way of illustrating the points we are making.

**Response to Reviewer 1 Comments**

Reviewer comment:

The manuscript is well-written and presents interesting data, particularly the author defined term 'growth yield'. Arguably, some conclusions are anticipated (based on physiochemical understanding) and the work appears limited to the alpha-pinene + O3 system in the author's flow tube. This subsequently raises questions regarding scientific significance. My main comments are: 1) model over-simplification, and 2) scientific significance.
The model is based on six alpha-pinene oxidation products, each with a specified volatility bin. Given that many tens to hundreds of products are detected in α-pinene SOA in flow tube experiments, I question the representativeness of the model and lack of supporting measurements. Further, some physiochemical properties do not appear to be considered (or at least, are unclear in the text), e.g. gas-phase dimer formation, volatilisation and gas-particle partitioning following in-particle compositional change. Moreover, the model appears heavily weighted to dimer formation. I suspect the authors other can clarify the above. However, while the authors note that the model is described elsewhere, further information must be included in this manuscript. Please address: i) if the selected model parameters (largely based on literature observations) are 'applicable' to this work (e.g. similar experimental conditions), ii) why few oxidation products were selected, and iii) any potential limitations of the above points and the model. I note that some discussion has been included on the latter.

Author response:

We trust that the discussion we have provided above about the SOA simulations answer the reviewer concerns about model over-simplification and the selection of modeling parameters. Some specific changes to the revised manuscript not mentioned in our general response:  Our choice of restricting the simulation to a combination of surface- and volume- limited processes is discussed in the revised paragraph 2 of Section 2, including references from the literature.   Also in this paragraph, we included new discussion of a particularly important physicochemical property – hindered diffusion in the particle phase.

Reviewer comment:

It is difficult to assess the scientific significance of the work. The authors make little reference to other particle growth models or prior observations (either modelled, physiochemical or compositional) to critically evaluate their data. Further, the model itself appears restricted to the alpha-pinene + O3 system in the author's flow tube (very specific). How representative is the data? Have the authors investigated kinetic limitations in other VOC systems or flow tubes, or can literature be used to support the data presented here?

Author response:

Hopefully, our general response addresses this comment. The goal of the manuscript is not to model actual flow tube data, but to effectively test the interpretive method for assessing flow tube data when the chemical details of particle growth are unknown. The interpretive method can be applied to any VOC system studied with the flow tube. We have added references to models of α-pinene SOA formation to paragraph 2 of Section 2, though this list isn't exhaustive since we're not trying to develop an α-pinene SOA formation model.

Reviewer comment:

Further, the manuscript conclusions did not convey the significance of the work and how the model or data may be of use to the scientific community. Do the authors intend to share this model to aid others in the understanding of particle growth in flow tubes (assuming this is possible)? Or will the model be used solely by the authors to characterise other VOC systems to provide physicochemical insights (with modifications to the model parameters, I suspect)? Most importantly, can the authors demonstrate that these insights are representative and not just applicable to their flow tube?

Author response:

Hopefully our general response has addressed these issues. The significance of our work is in the robustness of the interpretive method as shown by revised Figure 5. This approach can be used with any VOC system, and the GFs determined with our flow tube experiments should be directly relatable to those determined by another laboratory under similar conditions.

Reviewer comment:

Overall, I believe the manuscript is within the scope of AMT and that the work presented is suitable for publication following **major revisions** in the presentation of the manuscript. I suggest the authors reduce the technicality (where possible) and length of the text (making use of a supplement), reduce the number of figures in the main manuscript to approximately five at most (currently includes ten) and address the above comments to strengthen the scientific significance and conclusions. Finally, noting the title, "modelling ultrafine particle growth based on flow tube reactor measurements". Please include the measurement data and a brief description of the flow tube in the manuscript. Some specific comments are shown below (not an exhaustive list).

Author response:

Thanks very much for the suggestion to reduce the number of figures. As it worked out, five is just the right number. As indicated in our general repsonse, the "experimental" data in this manuscript, so to speak, are the SOA simulations in Sections 2-4. We changed the manuscript title to: "*modelling ultrafine particle growth in a flow tube reactor*" to remove the 'measurement' concern of the reviewer comment.

Author responses to specific comments from the reviewer:

Line 6: Please rephrase this sentence. The sentence reads that higher mixing ratios are used because of significant particle growth in flow tube experiments. Rather, higher mixing ratios are generally used because of the short residence time in flow tubes.

Response: Our revised abstract does not include this sentence.

Line 92: "…enter the particle phase and stay there…". The use of "stay there" reads as indefinitely. Include "over the investigated time frame" or similar.

Response: Change has been made (line 81 in the revised manuscript).  We also included this phrase elsewhere in the manuscript for consistence.

Figure 3: Please include the growth time duration in the caption.

Response:  Figure 3 of the original manuscript is no longer included in the revision.

**Response to Reviewer 2 Comments:**

Reviewer comment:

This manuscript describes a modeling study of aerosol growth in a flow tube reactor. The authors frame the discussion in terms of a new quantity they have introduced: the growth yield, which is a measure of the fraction of molecules generated in a flow tube reaction that contribute to particle growth. It can depend on various parameters such as rates of reaction, reactant concentrations, particle size and concentration, particle-phase reactions, etc. Here they show how these different parameters affect the growth yield, and describe how one can use that to extract information from measurements of particle size change in a flow tube reaction.
I find the concept and its value a little difficult to grasp, but as it becomes more widely used I expect that to become more apparent. Some of the difficulty may have arisen because the model was not applied to any measurements in this paper, since that has been done previously. The presentation is well done and the authors provide a clear discussion of the results and interpretation. My primary questions have to do with how many different ways one can accurately model a rather limited set of experimental results. I worry that because of the complexity of most reaction systems that it is difficult to constrain the model. I state some of these concerns below, but there are certainly other issues one would wonder about capturing in a model. I think the approach is novel, and may find use in the aerosol community, and so is appropriate for publication in AMT after the following minor comments are addressed.

Author response:

We hope our general response clarifies our use of a simplified SOA formation model, especially that we are not trying to use the simulation to accurately describe SOA formation for α-pinene or any other specific VOC precursor, but rather to test the interpretive method.

The value of the growth factor (yield) is discussed at the end of the second paragraph in the revised Section 5: "*An important point to note is that the wall loss and condensation sink terms in Eq. 9 are what make it difficult to simply compare outlet-inlet particle diameter changes from one flow tube experiment to the next, since the magnitudes of these terms affect how much growth is observed and they are not necessarily constant from experiment to experiment. Growth factor overcomes this problem and is specific to the VOC system being studied.*"

Author responses to specific reviewer comments:

1. Why don't the more volatile SVOCs contribute to the growth yield? Even though they are more volatile, they have higher concentrations and because dimer formation is fast it seems like enough should partition into particles that they can form non-volatile dimers.

   Response: For the specific simulation conditions we use, we find that more volatile SVOCs don't contribute substantially to the growth we calculate, so we have not used them in this manuscript. Of course, it is conceivable that more volatile SVOCs contribute to particle phase chemistry in actual systems. Again, our simulations aren't meant to reproduce SOA formation for any specific VOC system, but rather to test the interpretive method. Here is the text we added to the last paragraph of Section 2.3: "*In principle, this term represents all possible combinations of NVOC and SVOCi molecules to form a DIMER However, the simulations shown in this study include just one specific DIMER formation reaction – two SVOC$_0$ molecules reacting with each other. Simulations including the full range of DIMER formation reactions have been performed, but they add very little to calculated diameter growth (most of the growth is due to SVOC$_0$ only) or to the time dependence of GF in the flow tube. Therefore, in the interest of simplicity for discussion of this work, only the SVOC0 dimerization simulations are shown.*"

2. How well are the rates of loss of products to the walls understood, and how does this impact the modeling? Is it irreversible or reversible, and won't this also depend on how much organic or water is on the walls?

   Response: The rate of wall loss is very important in an actual experiment, and certainly depends on a variety of factors. It is not really relevant to this manuscript because we are just simulating SOA growth, and in principle we could choose any value. We include it in the Eqs. 1 and 9 along with the discussion of Eq. 9 (see initial author response to this reviewer) to help drive home the importance of the growth factor concept.

3. What happens to the model when dimer formation is treated as reversible instead of irreversible?

   Response: Certainly, reversible DIMER formation could affect (most likely decrease if dissociation is fast enough) the amount of growth calculated in the simulation. However, the goal of the manuscript is not to develop a detailed SOA formation model, but to simulate SOA formation for the purpose of testing the interpretive model. Hopefully, our general response provides clarity on this point. We also added the following text to Section 2 paragraph 2 (line 115): "*…they do not include, for example, reversible dimer formation or the possibility of hindered diffusion…*"

4. Isn't water likely to affect the dimer formation process, for instance completing with hydroperoxides in dimer forming reactions with aldehydes, or shifting equilibria of dimer formation by dehydration reactions?

   Response:  We briefly discuss hydroperoxides and their reactivity in paragraph 3 of the introduction and in the first paragraph of Section 2.3.   Again, we hope our general response provides some clarity to the reviewer's query – the goal of the manuscript is to use simulated SOA formation to test the interpretive method, not to develop a detailed model for SOA formation.

5. It is thought than aerosol particles often exist as phase separated organic/aqueous solutions. Can the model capture this?

   Response:  In the simulations, we do not include phase behavior beyond distinguishing liquid-like (total of ALW and organics) from solid phases, nor do we consider the possibility of different chemical reactions in different composition liquids, see the first paragraph of Section 4.3: "*SVOC partitioning and DIMER formation are assumed to be independent of whether the reactive phase is aqueous, organic, or a combination of the two.  Of course, in experimental systems this probably is not the case, but the purpose of these simulations is to explore the effect of total reactive volume, not phase-dependent chemistry.*"

   Author responses to technical comments:

   1. Line 157: Should be "recursively".

      Response:  Change has been made (line 174).

   2. Line 190, Equation 7: The text in parentheses is not clear.

      Response:  The text has been revised for clarity (Equation 7, line 209).